# “Kept in Check”: Representations and Feelings of Social and Health Professionals Facing Intimate Partner Violence (IPV)

**DOI:** 10.3390/ijerph17217910

**Published:** 2020-10-28

**Authors:** Immacolata Di Napoli, Stefania Carnevale, Ciro Esposito, Roberta Block, Caterina Arcidiacono, Fortuna Procentese

**Affiliations:** Department of Humanities, University of Naples “Federico II”, 80133 Naples, Italy; immacolata.dinapoli@unina.it (I.D.N.); stefania.carnevale@unina.it (S.C.); ciro.esposito5@unina.it (C.E.); roberta.block81@gmail.com (R.B.); caterina.arcidiacono@unina.it (C.A.)

**Keywords:** IPV (intimate partner violence), gender-based violence, social and health professionals’ feelings and thoughts towards IPV, reflexivity, specialized training, treatments

## Abstract

Social and health professionals facing gender-based violence in Intimate Partner Violence (IPV) express feelings and thoughts closely connected to their place of work and the users of their services. However, research on professionals’ reflexivity and their implications has not been closely investigated. Therefore, this article will describe representations of IPV among social and health professionals facing gender-based violence as well as their personal feelings in accomplishing their job. Fifty interviews with health and social professionals were analyzed using grounded theory methodology supported by Atlas.ti 8.4. Five macrocategories will describe this phenomenon, leading to the final explicative core category that summarizes professionals’ attitudes toward it. Being “kept in check” among partners, partners and families, services, and institutional duties is the core category that best expressed their feelings. Therefore, implications for services and training will be further discussed.

## 1. Introduction

Violence in intimate relationships, or Intimate Partner Violence (IPV), refers to, according to the World Health Organization [1], “any behavior within an intimate relationship that causes physical, psychological or sexual harm to those in the relationship”. IPV refers to violence between two people involved in an intimate relationship, currently or in the past, and it is a transversal phenomenon, present in all cultures and societies [2,3,4,5]. 

In literature, the term IPV has recently been preferred to the term “domestic violence” for emphasizing “the violence that occurs by partners or ex partners, therefore within a love/sexual relationship, whatever the level of intensity and regardless of coexistence” [6] (p. 17). Another name for this form of violence is “trust” violence, a term coined by the Italian sociologist Ventimiglia [7] to define violence that occurs within trust relationships in which, by natural inclination, people are led to rely on someone, trusting them, to feel safe and protected from external dangers. Therefore, it refers to the type of violence that is undermined precisely in those relationships where the love for the other allows a relationship of trust and a feeling of protection that makes it difficult to recognize and accept violence. They are intimate relationships in which one relies on the other as a “safe haven”, but then what happens is that you have to defend yourself from the very same person you trusted. Specifically, it is rarely an episodic or exceptional explosion, but rather it is characterized by its progressive and exponential escalation, being structured and not exceptional [8]. Another important aspect is the “transversality of violence”: a feature that, according to Barbagli and Saraceno [9], explains how this type of violence is not attributable to specific social groups and, at the same time, it is not superimposable to social marginality and deviance. IPV does not occur in a specific type of relationship or type of person; there are no typical traits of the abused woman or the man who perpetrates violence, but it can be reiterated at any level of social position and it can affect all kind of people. Finally, an additional element is its continuity over time [7]. In fact, the author talks about "a relational dimension of ordinary violence in everyday life". In its forms, there is a destructive commuting that alternates various types of violence: it goes from verbal violence, to economic violence, from physical abuse to psychological violence, to the total silence of the man, which is also a form of violence. With silence we communicate to the other that he/she is “nothing” or “nonexistent”. Nonresponses "cancel out" the other by not recognizing him/her.

Several studies on the prevalence of IPV have shown that both men and women can be victims [10,11]. However, from data on the incidence of the phenomenon it is clear that the latter are more likely to experience partner violence [12,13,14,15] and to be subjected to sexual violence [16,17]. In addition, a systematic review on IPV found that violence is prevalent in adolescence and in young adult women [5].

Moreover, statistics show the strong predominance of femicides among crimes that occur in family contexts [18,19,20]. Women victims of violence report both serious physical consequences [21,22], as well as post-traumatic stress disorder effects [23,24,25].

The aggressive and coercive physical, psychological, and sexual behavior that characterizes the violent intimate relationship is often generated and justified by gender dynamics involving a culturally shared patriarchal logic. This conceives the man as the holder of power within a relationship, and his behavior is aimed at controlling the partner [26,27,28,29,30,31].

For many years, the problems deriving from the socio-cultural attitude wherein the main objective has been to maintain the integrity of the family have been underestimated, if not actually ignored [30,32,33]. Many epidemiological studies, especially over the last few years, have been conducted in order to investigate the diffusion of this phenomenon. In 2013, the WHO produced a systematic review of 141 studies conducted in 81 countries. The results demonstrated that approximately 35% of the women examined had experienced, over their lifetime, physical and/or sexual violence on the part of their partner, or sexual violence inflicted by a stranger [34,35].

Abramsky et al. [36] suggested that a protective factor against IPV risk is completing secondary education, whereas primary education alone fails to confer similar benefits.

### 1.1. Intimate Partner Violence (IPV): The Violent Relationship and Its Consequences

IPV can occur in different forms: physical, sexual, psychological/emotional, and economical violence [2] and it is part of that “possible violence” that can take place within the home, involving not only the couples, but also the members of the family at large and the children in their role of witnesses to violence [37,38].

Literature has amply analyzed the salient aspects of the psychological process underlying the relational dynamics within which violence in intimate relationships takes place [39,40,41,42] and the consequences on the victims continually exposed to violence [43,44,45,46,47,48].

One of the first studies that examined the consequences of violence on women from a feminist perspective dates back to Lenore Walker [44,49,50]. In “The Battered Woman Syndrome” [49] the author identified the symptomatology of the woman victim of physical, sexual, and/or psychological violence, identifiable through six criteria: intrusive memories of the traumatic events, high levels of anxiety, avoidance behavior, disrupted interpersonal relationships, body image distortion, and sexual intimacy issues [44,49]. Walker identified, during her professional and research experience and fieldwork, what she herself called the “cycle of violence”: a particular evolution that characterizes the violent relationship. The abusive behavior is never limited to a single episode but it is repeated cyclically through a predictable cycle of three phases: the tension-building phase, the acute battering incident, and the begging for forgiveness that leads to making up. It is this last phase that keeps the woman “bonded” to the man. This return of man to loving and submissive ways and his request for forgiveness are often accompanied by promises of change and interact with a whole series of psychological mechanisms and inescapable circumstances that keep the bond in place (e.g., economic and emotional dependence). Therefore, the violent relationship is carried on. Once the cycle ends, it soon repeats itself, and peaceful times become shorter and shorter. 

As far as the perpetrators of violence are concerned, in a psycho-dynamic approach Mizen [51] supports the idea that men’s violence towards women is a pathological variation of aggression. These individuals are, in fact, incapable of elaborating their emotional experience, and create a relationship with the woman founded on the pattern of projection, and live with the worry of separation from her as “internal nullification”, an unthinkable threat, a nullification of his own identity built on her. These are men who are incapable of thinking and elaborating their emotions. For this reason, they feel anger towards the lack of engagement of the woman and act upon it. The woman is the depositary for parts of their Self; he can not separate himself from them, nor can he disengage from their structured identity [39,51]. This psychodynamic reading does not cancel the perpetrator’s responsibility, but captures a particular relational and psychological mechanism (projective identification) and the profound experience that derives from it in the moment of separation, in which the inability to think about the emotions leads them to act them out.

There is a whole social and cultural aspect of transmission and adhesion to a model of masculinity linked to the possession of the woman, to control over her and to the supremacy of the man. Therefore, on a cognitive-behavioral level the man forms mental scripts for which certain behaviors are part of a socially reiterated and shared "normality” [52].

Therefore, IPV has not only unconscious aspects, but is largely intentional and is defined as a choice, allowing the perpetrator, therefore, to take the alternative path of nonviolence.

In this vein, the gender stereotypes underlying male violence against women, the reasons put forward to justify it, its symbolic function, and finally, the benefits that men could derive from perpetuating violence must always be evaluated as behavioral and educational aspects of the phenomenon [52].

Thus, individual, but also relational, cultural, and social factors intervene in this “relational gridlock”. 

Partners in IPV relationships often share and reiterate, in fact, gender stereotypes that, often, in a patriarchal logic, establish the characteristics attributed to the male and the female, but also the expectations connected to male and female behavior [53].

In this view, the violent episodes are legitimized within the adherence to the traditional gender standards and roles that attribute virile masculinity to the man, who recognizes himself in power, force, and control. The same gender stereotypes that reiterate socially shared roles and functions within the family are often introjected by the woman and acted out in couple relationships. The woman’s adherence to the female gender role, intertwined with her family and personal history, her experiences, her attachment style, could justify the woman remaining in the abusive relationship and her high tolerance for physical, sexual, and psychological emotional abuse [54]. 

While there is extensive literature on the characteristics and consequences of IPV, the interest towards the representations and emotions of professionals who work with violence between partners is scarce [55], as well as the expression of their reflexivity concerning this topic. The literature [56,57,58] has highlighted the connection between psychological consequences such as states of depression, post-traumatic stress disorder, and suicide attempts related to IPV. 

Campbell [59] further examined the consequences of IPV on women’s health, identifying physical and mental disorders. The most serious risk for these victims of violence is also the occurrence of a series of somatic and psychological health problems [34], and this highlights the need for adequate training for health professionals so that they can recognize symptoms and consequently identify IPV cases [60]. As described in the recent literature review [61], it is very important for primary care professionals to adopt a biopsychosocial approach in order to offer emotional support and information about resources to IPV victims.

Particularly, studies reveal the invisibility of IPV among healthcare and social service professionals [62]. Further studies reveal that the obstacles to the adequate management of IPV lie in the preconceptions surrounding the factors that determine violence in couples [63], and a lack of appropriate training [64].

Therefore, we set up an ambitious project to extend knowledge about the professionals and services dealing with IPV. In a first paper we analyzed professionals’ representations of children witnessing violence in the family [38], then we did a thematic analysis of professionals’ views of needs and resources of the different social and health services aimed at fighting IPV [65]. Lastly, in a companion article we presented the representations of professionals concerning perpetrators and intervention strategies to deal with this phenomenon [66]. Here, in this article, the aim is specifically to describe the professionals’ attitudes and resources in their management of IPV dealing with both perpetrator and victim and their reciprocal interactions, considering also that the relationship among professionals and victim and perpetrator assumes specific characteristics [67]; in fact, our research on professionals dealing with women victims of gender-based violence has enabled us to detect their specific vision of IPV and their reflectivity and positionality in treating such cases, with the aim to better understand how they feel and interact with their clients and service goals as well as circumstances and feelings that may prevent social, psychological, and cultural support and qualified action of services [68]. 

### 1.2. Objectives

The aim of this study, carried out within the context of gender-based violence, is therefore to analyze the professionals’ experiences and portrayals of the IPV phenomenon and the perception of their own position regarding the relational dynamics between victim and perpetrator. The choice of qualitative method was considered to be the most appropriate. This approach, in fact, allows us to better understand the participants’ experiences and their interior world, to discover their representations and meanings attributed and built around the phenomenon, and to probe into the individual, relational, and social aspects related to their environment.

## 2. Materials and Methods

In order to pursue the objectives, open interviews were conducted with operators who work in the management of couples, men, women, and/or children involved in dynamics concerning IPV. The interviews were audio-recorded, transcribed, and analyzed through the Grounded Theory Methodology (GTM) [69] (Bryant). The GTM was chosen because it allows the researcher to explore data collected and to create a new theory through a process of multiple coding activities. “GTM offers a specific combination and implementation of coding, conceptualizing, abstracting, and theorizing” [69] (p.19) (Bryant), leading to new theoretical models. 

### 2.1. Participants

There were 50 participants in the study—45 females and 5 males. They all had experience in the field of domestic violence prevention and management. They were aged 27–70. There were both volunteers and professionals (with a range of service in the field ranging from 1 to 45 years, SD. 12.35; mean: 45.56). They were selected among cultural, political, and social workers who are involved with domestic abuse, in different roles and who work in services located in Naples. Specifically, they were psychologists, psychotherapists, social workers, nurses, and magistrates hailing from various services in central Naples and its outskirts. Of the participants, 54% had worked with women victims of violence or had participated in the planning and execution of ad hoc projects. Of the participants, 46% had worked in person with the perpetrator. The participants were selected through a theoretical purposive sampling [70]. Table 1 shows further details about participants.

The theoretical sampling consisted of identifying participants by following the indications given by the process of analysis. The participants in the study were contacted by telephone to make appointments. The interviews were held at their place of employment or at the “Federico II” University premises. They were held in quiet, reserved environments and lasted from 30 minutes to 2 hours each, with an average length of 50 minutes. The research team were committed to setting up times and days that were compatible with the needs of the interviewees. All the interviewees gave their consent and authorized the audio recording and the use of data for research purposes. Ethical statement, approval number: 15b/2019.

### 2.2. Instruments

The instrument for research used was the open interview, according to principles by Legewie [71], Schütze [72], and Arcidiacono [73]. This method lets the interviewee be free to describe their emotions and thoughts and does not call for specific predetermined questions. The interviewer is able to pick up on the content provided in the interview, being knowledgeable about the topic discussed, and the general and specific aims of the study. Each of the selected interviewers are skilled in this as well as being extremely competent in carrying out interviews for qualitative research grounded on a psychology basis. 

The topics of the interview, identified with the aim of comparing the study to the most recent IPV literature, were as follows: a. The representation of the IPV phenomenonb. The representation of the woman victim of violencec. The representation of the perpetratord. The intervention procedures (already used or suggested and not yet put into action)e. The influence of gender of the male and female professionals in the therapeutic relationship

The recordings of the interviews were transcribed verbatim and analyzed—according to Grounded Theory Methodology [69]—with IT support for text analysis. In this study, ATLAS.ti software was used (Scientific Software Development GmbH, Berlin, Germany). 

The coding procedure was carried out in three phases by a team of researchers with a range of experience in the field of IPV. In the first initial coding phase (open coding), we proceeded to fragment the text material into quotations that were part of the texts of varying length. They were grouped into some codes assigned to them, corresponding to a synthesis of their specific sense. These codes were then grouped into wider categories, and relationships between each of them were identified through the axial coding process. During the last phase, that of theoretical coding, these categories were articulated into deeper thematic macrocategories of meaning. In the coding procedure, a consensual validation mode among researchers was put into place to allow the assigning and grouping of codes during the reading and interpretations of the texts collected. 

During the coding phases, the group of researchers constantly consulted each other in order to identify categories, which included common codes, and outline new conceptual categories based on the differences. The comparison process among the researchers allowed the definition of the core category as a central and explanatory category of the management and position taken on IPV by the interviewed professionals. It represents a more abstract conceptualization achieved through the mechanism of the group insight.

## 3. Results

The codes assigned as a whole were grouped into several categories and macrocategories, but here we are analyzing only 135 codes grouped into 5 macrocategories, reporting those specifically referring to the IPV and the professionals’ interactions with victim and perpetrators and their professional actions: 

3.1.Representation of the effects of violence on the woman;3.2.Representation of the woman victim of violence and her relational style;3.3.Representation of the man inflicting violence and of his relational style;3.4.Representation of the elements which keep the violent relationship alive;3.5.Representation of the relationships between professionals, service providers, and users.

### 3.1. Representation of the Effects of Violence on the Woman

This macrocategory includes the categories involving what the professionals report as the effects of the violence.

*Self-denigration of the woman*. Often self-blame for the violence endured is reported in the woman, as well as difficulties in parenting roles, sometimes accompanied by a sense of guilt towards the children

(The woman victim of violence) “*is a woman who doesn’t believe in herself, and doesn’t believe in her potential. She is a very fragile woman, with a lot of guilt, who tends to feel guilty about everything that happens, and so she keeps it all inside, she isolates herself on a social level because of her sense of shame*”.(F, 31, Psychologist and Psychotherapist, workplace: Perpetrators’ shelter OLV—Oltre La Violenza)

*Difficulty in the parenting role*. Professionals interviewed also described the women victims of violence as neglectful mothers: sucked into the violence endured, trapped in the dynamics of a controlling and dysfunctional relationship. They find it difficult to defend themselves and, consequently, to protect their children, whose needs they often do not see, because they are often acting in “emergency mode”.

*“In the same way you can’t protect yourself, you can’t protect your children. Unfortunately, that’s it. I think there is a lack, in the person, of the ability to protect themselves…how can you then protect the young ones?”*.(F, 39, Psychologist and Psychotherapist, Workplace: service combatting gender-based violence against women CAV—Centro Anti-Violenza)

On both a physical and a psychological level, they say that the effects described are recognizable as symptoms of post-traumatic stress disorder [74,75,76].

### 3.2. Representation of the Woman Victim of Violence and her Relational Style 

The second macrocategory includes codes referring to the way in which professionals and workers portray the woman victim of violence and her relational attachment style.

*Dependency.* The words of the interviewees highlight above all the image of a dependent woman, with low self-esteem and autonomy limited to certain contexts (e.g., taking care of the children), or completely lacking.

*“Often they are women who have problems with emotional dependence…so they have low self-esteem, very low...growing up with the idea that the man is the one who decides, who has a certain control, a certain power…”*.(M, 49, Sociologist, workplace: Campania Region office)

*Denial.* The professionals describe this when they find themselves facing women who appear, in their view, to be unaware of the violent situations they are experiencing:

*“so maybe they don’t turn to a psychology service, or they do, but with great ambivalence, so then they decide not to go on with the process…”*.(F, 42, Lawyer, workplace: Association Movimento Forense)

One of the first steps to take, as professionals, is actually to help them recognize the violence, to see it and to name it:

*“so, when we run into some cases of violence and abuse that’s not recognized by the victims, that’s the support path to take, not only to allow the abuse to be seen, but also to recognize it as such, this is the first aspect”*.(M, 46, Executive officer, workplace: Campania Region office for Equal Opportunities and Human Rights)

*Savior complex.* The attitude of women victims of violence towards their partner, defined by the professionals as “Florence Nightingale syndrome”, is often connected to a sense of omnipotence with which these women face the dysfunctional relationship and its consequences:

*“There’s always this thing “I’ll save you” that still works...even in us, with backgrounds in feminism, …that ‘savior’ is always there …lying in wait, so to say”*.(F, 60, Sociologist, workplace: service combatting gender-based violence against women CAV)

### 3.3. Representation of the Man Inflicting Violence and of his Relational Style 

In this macrocategory we find the traits and relational style of the man who is the perpetrator of violence, according to the interviewees.

*Manipulation and acting-out.* The interviewers attribute to the perpetrator a relational style characterized by a marked tendency towards manipulation and the inability to engage with his own emotions or share them.

*“They are very seductive men, tending towards being manipulative and fragile”*.(F, 44, Social Worker, Service for Youth and Families)

*Fragility*. As one psychologist states: “*They are fragile and disconnected men, because being violent doesn’t mean being strong… I’ve always met very fragile men, so I think that the man who is a perpetrator of violence is often a person who, because he is so fragile, uses physical force to affirm himself over the other and to feel recognized*” (F, 66, Psychologist and Psychotherapist; workplace: Perpetrators’ shelter OLV).

Although the interviewees recognize the fragility of the perpetrator in their use of violence to affirm themselves, at the same time they have to succeed in not being “taken” by that feeling, and to pursue a path of awareness and change with them.

### 3.4. Representation of the Elements which Keep the Violent Relationship Alive 

Professionals describe the bond between partners (in IPV cases) as fusional: they often talk about symbiotic relationships and the way in which each member of the couple tends to structure their identity on that of the other. 

*Relational interlocking.* The interviewees describe the relationship between victim and perpetrator as being based on strong complicity:

*“… if a strong bond isn’t created then one doesn’t stay… I believe that we have to understand how far the complicity reaches”*.(M, 70, Psychologist and Psychotherapist, Judiciary service)

The interviewees highlight that the bond between the two partners is characterized by strong co-dependence, and, when the woman sees her life and her children’s lives at risk, she tries to “break” it.

*“Often people who suffer from attachment dependency choose a partner who is also problematic… motivated by the intention of saving the other from his problems”*.(M, 70, Psychologist and Psychotherapist, workplace: Judiciary service)

*The reciprocal omnipotence and destructivity of the violent couple.* Furthermore, the interviewers add that the relationship dynamic is characterized by a sense of reciprocal omnipotence and deep destructivity. According to professionals, this starts to appear through denigration (sometimes reciprocal) and aggressive verbal interaction. According to some, it can affect all members of the family unit, even the children. In addition, it does not occur outside the walls of the home. The professionals talk about a “simulated normality” to outside eyes, on the part of both members of the couple.

*“The woman feels like she’s really able to tolerate the violent man, so there’s a perverse dynamic related to the omnipotence… she is not aware of her feelings of omnipotence, so they are hard to access through awareness, but they are very dangerous, inducing the woman to tolerate every sort of violence”*.(F, 31, Psychologist and Psychotherapist, workplace: Perpetrators’ shelter OLV)

*“Because there’s no more pleasure, no more serenity, no more equilibrium, and then there’s the risk, indeed, not only of losing one’s life, the peak being femicide, but the loss of one’s mental wellbeing, because it’s a destructivity that actually becomes a threat to health and mental stability”*.(F, 39, Psychologist, service combatting gender-based violence against women CAV)

These elements explain why for the professionals and services it is so hard to promote changes in the reciprocal relationships among partners and support the woman in her ending the relationship as well as the man in tolerating the estrangement of his partner. These specific features of IPV require the professional to present a high level of patience and competence and specific training as well as service monitoring and evaluation. The development of further intervention strategies dealing with different shelters and services is also required.

### 3.5. The Representation of The Relationships between Professionals, Service Providers, and Users

The main issue for professionals is to deal and interact with the “power game” within the couple and this dimension requires specific training:*Holding the power.* Power over women is reported to be one of the main dynamics on which men act.

*“In these conflictual dynamics which then result in violence, power is activated…but as a perpetrator explained to me: ‘it is the power that I try to hold over others and the pleasure I get in holding this power, probably often linked to the fact that I, instead, am powerless over some of my emotions, some of my experiences’…”*.(F, 66, Psychologist and Psychotherapist, workplace: Perpetrators’ shelter OLV)

*Taking over the other.* Power and taking over the other leads to the man’s extreme control over the woman, but as described in an interview, in virtue of the extreme co-dependency of the couple, the woman also may become controlling. “*The dynamics of control is implied, and it doesn’t always and only involve the partner who is violent towards the victim, because in a certain way even the victims have control or at least manipulate, … eh?”* (M, 31, Psychologist, workplace: private practice).

In this power game within the couple, the professional risks not finding room to intervene and is forced to powerlessly observe its atrocious effects. This dimension brings out how important are the professional skills of personnel and opens the discussion about their training and specific competencies.

In regards to the professionals’ action, the focus is also on the gender of the professional involved with the user (female or male).

*Active listening and lack of judgement, regardless of gender.* Almost all the interviewees highlighted the difficulties, but also the absolute need for always maintaining a nonjudgmental attitude and for explorative listening, whether it be in the management of a woman victim of rape or the perpetrator.

*“as a woman, mother, wife, I tend to see the tormenter, the monster… but, in any case from a professional standpoint we have consider that there is a human being with difficulties and experiences”*.(F, 57, Psychologist, workplace: family services)

*Gender role in the relationship with the user.* This regards the importance of the gender of the professional involved with the user. All the participants said it was fundamental that women victims of violence were welcomed and managed by female professionals (not only because this is required by law). As far as the relationship between professional and male perpetrator is concerned, pros and cons to both have been brought to light.

*“it’s true that professionalism should not have a gender and all that, but for this type of problem it’s important, I think…there are women that clearly say ‘Just as well you’re a woman!’, so I think it is of great influence”*.(F, 35, Psychologist and Psychotherapist, workplace: service fighting gender-based violence against women CAV)

*“Violent men, when there’s a woman facing them, tend to undermine…for example, I’ve always been told ‘Yeah well, she’s allied with women… she’s an ally to women, surely she can’t understand me, she doesn’t defend me’!”*.(F, 66, Psychologist and Psychotherapist, workplace. Perpetrators’ shelter OLV)

*“In the male mentality there’s the idea that ‘I’m kind of a son’, no? ‘I’m my mother’s son, then I’m my wife’s son‘, no? I mean, it becomes a bit like that. And…. the fact that I find a woman in front to me, it represents the same pattern. I mean, I can fall into that thing, so much so that I... at least from the descriptions I’ve read, from what they tell me…’If I did it, I did it because she provoked me!’, no? It’s a bit like a child justifying himself to his mother”*.(F, 61, Psychologist and Psychotherapist, workplace: Perpetrators’ shelter OLV)

*“…the perpetrators aren’t greatly introspective people as you can imagine, on the contrary, they’re people with a tendency to act out to the max and have an attitude that’s almost saying, for example, at the moment I’m thinking of a person whose session was quite, not very simple, because he acted in an aggressive and arrogant manner”*.(F, 66, Psychologist and Psychotherapist, workplace: Perpetrators’ shelter OLV)

The gender of the professional is then an element that can contribute to being kept in check and of which we need to be fully aware. Woman to woman brings about easier reciprocal identification, and sometimes more difficulties in envisioning ways out. Woman to man can bring about a refusal that hinders communication. Finally, a male professional to a woman can induce less sensitivity to the requests expressed and with a male user a collusion in a male chauvinist view of the relationship with the partner.

The perpetrators must be given a way to manage their emotions and aggression, in order to understand the origins of their anger, frustration, and insecurity, and then to support their overcome [65,77]. A female professional who is managing a perpetrator must not feel “disloyal” towards the female victim, but be aware that work with the man is necessary. Effective intervention for support is needed for the many situations that arise during the path from reporting the violence to overcoming it [32].

### 3.6. Core Category and Network 

The grounded theory methodology allows the researchers to think about the issue that is best featured in the texts under examination. Reading together and discussing the quotations, codes, and categories leads to a second level in explaining the concept that brings the researchers to a deeper understanding of the topic. “Kept in check”, with this expression we intend to describe the effect produced on the professional in dealing with the perpetrators and victims of intimate partner violence.

The participants in the study talk about “manipulation” on several levels: on the part of the perpetrator, firstly towards his partner, but also towards the professionals working in social and health services, in the attempt to appear to be a victim of his partner’s provocation, not the executioner, often underestimating the seriousness of the violence inflicted. Often, in fact, the man does not agree with the assignment of the professional, and only sticks to the requests of the service to then be able, at a later time, to turn the tables in his favor (one interviewee reported the following words of the man often said to women: “See? I also went to psychologists, you’re the problem, you’re the one that provokes me, you’re the crazy one”. At the same time, according to the words of the interviewees we can infer a feeling of being manipulated by the woman: at the time of turning to services, she seems to be asking to “fix the broken object” so that they can “take it back home”. It’s almost like she’s saying to the professional “I‘ve done everything I could, now you take care of it”.

The manipulation within the dysfunctional relationship reflects onto the relationship with the service. In reading the interviews what emerges is how the professionals find it difficult to find in women the resources with which to construct a process of change. The risk that emerges is that of not succeeding to lever the emotions and thoughts that could trigger a change. The challenge, then, is to not remain stuck in the suffering and to manage to undertake the difficult path towards emerging from the violence.

The professional sees the woman as powerless and delegating (“*Look what he does to me! Look what he’s capable of!*” she seems to be saying) without recognizing any potential and resources, standing firm in her view of reality and longing for revenge (“*I’ll make you pay for this*”) through the professional. What happens is the professional feels controlled by the woman when, for example, the woman calls to know if her partner has been turning up at his appointments. The professional also ends up being an unintentional go-between for users of the service and judicial authorities. Moreover, as we acquire from legal procedures [78], professionals, especially in their role of judiciary consultants, need to understand how the violence suffered by the partner affects women’s wellbeing and their mothering competence. Therefore, even just the escape from the violent relationship will bring advantages to the children and consequently, professionals have to consider that they are not in front of a “bad mother”, but a woman who, for the good of her children, is summoning up the courage to take her life back into her own hands and break the addiction to her partner.

In the interviews the professionals describe finding themselves in a paralyzing deadlock. The situation is presented as unchangeable, but at the same time there is an awareness that the request is for a saving transformative action. The professional enters into a scenario in which there are no resources, or there is no possibility of change; it is thus like being “kept in check”, as any type of possible intervention is considered lacking any chance of success, especially because of the difficulties involved in working in emergency mode. 

However, if the case worker yields to this view, they will not have any chance of intervening; at the same time, if they do not understand the scenario in which the man and the women lie, no regulation, measure, or proposal put into place will be strong enough to be activated. 

Indeed, all the participants reported that the activation of a reflective process on the therapeutic question "brought" by the users was fundamental; precisely, this allowed the researchers to reach the core category of the study. The experiences and analyses reported by professionals who work with women, those who work with couples, those who work with men, and those who have worked with both have allowed for the collection of a varied range of experiences. These allowed us to see the being kept in check and the attempted manipulation of the services.

The lack of specific training makes it difficult to acquire more awareness of their experiences on the part of the case worker, who is already immersed in a patriarchal culture.

Among the elements that keep the professional in check, one has to also consider the unidirectionality of the interventions offered by antiviolence services and those for the treatment of the perpetrators, as well as the lack of case management of the latter in general services. A lack of measures of empowerment for perpetrators and a shortage of them for women victims of violence add to reinforcing the stereotypes and dominant representations about relational dynamics [65].

The Figure 1 shows the Core Category and the network of relationships among the main concepts emerged from the analysis.

## 4. Discussion

This research does deal with some features of IPV, but its main goal is to describe the professionals’ point of view in dealing with such a topic: their potential resources and their “dark side”, namely, the results highlighted the decision-making and the circumstances of the professionals’ actions. The descriptions of the effects of IPV on the woman range, in the words of the interviewees, from symptoms comparable to those of post-traumatic stress disorder, to the effects of a sense of guilt, shame, and terror, as is also highlighted in several studies [79,80], all the way to difficulties in parenting roles [38].

The professionals consider the relational style of victims of IPV as characterized by a constant self-denigration. On the other hand, these women deny the dysfunctional character of the relationship they are living in, minimizing their partner’s violent behavior and assuming what the professionals define as a “Florence Nightingale attitude”, behind which lies, according to the professionals, a sense of salvific omnipotence. 

The perpetrator is described by interviewees as a narcissist with manipulative tendencies (even with the professionals themselves), incapable of thinking his emotions, fragile, insecure, and controlling within his relationship, in which he abuses his position of authority, as presented in the most recent literature [65,66,81].

As far as relational dynamics are concerned, roles, prejudices and gender stigma, shared and passed down in families and contexts have been highlighted [26,82]. So has the reproduction of violent behavior endured or observed as “normality of the phenomenon” [66,83].

These are aspects, confirmed in literature, and from a feminist perspective, [49,84] that center on the patriarchal dimension of gender-based violence. It is in this way the case worker is thrust into a whole intergenerational cultural system that does not recognize the characteristics of violence in the couple relationship. Systemic approaches emphasize the dimension of the rationality of the problem, but Autiero et al. [65] highlight the limits within the criminal judicial system. Several studies, however, of a psycho-dynamic and cognitive-behavioral slant, [51,85,86,87] highlight the influence of family and personal history in the acquisition of models of violence on the part of the perpetrators, as well as the presence of cognitive distortions and implicit theories that justify the violent actions. From this viewpoint, the influence of the family would be the cause of the actual violence, passed on from generation to generation through role scripts passed on from parents to children [88]. Then, the cultural and psychological dimension of the phenomenon is self-evident.

The specific element that characterizes this study is its focus on the way in which professionals and case workers perceive their own representations regarding the violent relationship and the way in which they experience them, from the moment in which the counselling/management relationship is activated. Literature does, in fact, tell us little about the experiences and symbolizations involved in the taking charge of IPV. 

## 5. Conclusions

The importance of the emotional, representational, symbolic, and relational dimensions has emerged as a central element in activating processes of reflexivity and awareness in the work of taking care of couples and individuals involved in IPV [89,90,91].

With the term reflexivity we refer to the awareness that the professional acquires concerning themselves and their role within the service, as well as the way in which their intervention is influenced by the characteristics of the users. The process of reflexivity allows professionals to recognize their position, that is the ability to understand their position as a professional in the web of relationships in which they are immersed and are part of [23,92,93]. 

Reflexivity leads to the achievement of “new levels of understanding” and “is a major skill needed in dealing with differences” [89] (p. 8); the aptitude to reflect on how our role and functions influence the others and their actions and how we are affected by them leads to our making our actions and emotions more responsible and thoughtful.

The interviews were considered by professionals as a time that allowed them to think about their work and, above all, their experience and their emotions about it.

The complexity of the IPV phenomenon requires an integrated and holistic view when working, in order to best understand and tackle it.

Exploring the experiences and representations of the professionals allows us to understand aspects of the phenomenon from a different viewpoint, and also to assess the strengths and weaknesses in preventative measures and therapeutic proposals geared towards the victims and the perpetrators [94,95], and especially the position held by the professional towards such a complex and pronounced dynamic. It is fundamental that professionals and service volunteers are trained in the understanding and management of their own emotional experiences in order to heighten and protect their own wellbeing, when they are in contact with users burdened with such complex requests. [96,97]. 

The limitation of this study is surely in the high rate of females among the participants, but this is, in reality, the prevailing component in services tackling violence. It would also be worthwhile examining the representation of IPV in a national or international population in order to attribute representations identified to a wider population that that of the Campania region. 

Although the feature of this study has been to examine the representations of IPV and its protagonists—victim and perpetrator—in the words of those working in the field, it would be necessary to integrate these representations of this phenomenon with the voice of the victims and perpetrators. In fact, only this extension of collected data can provide us with the full complexity of the phenomenon in the individuals’ experiences and the perceptions of the professionals working in the service, in the words of the users and the population as a whole.

The *Core Category* identified as a result of coding and categorizing, “professionals kept in check”, reveals itself to be the organizing concept of synthesis of the study. Actually, aside from underestimating and denying the inflicted or endured violence, what was observed was also a devaluation of the work performed in the case management and the objectives set by the services. Thus, “professionals kept in check” becomes a central factor that defines the perception that the professionals have of themselves when assigned an IPV case.

Finally, to confirm literature [77], the emerging elements bring to light the need for permanent professional development programs for the professionals, to allow them a more productive approach in the management of both the victims of violence and the perpetrator. 

## Figures and Tables

**Figure 1 ijerph-17-07910-f001:**
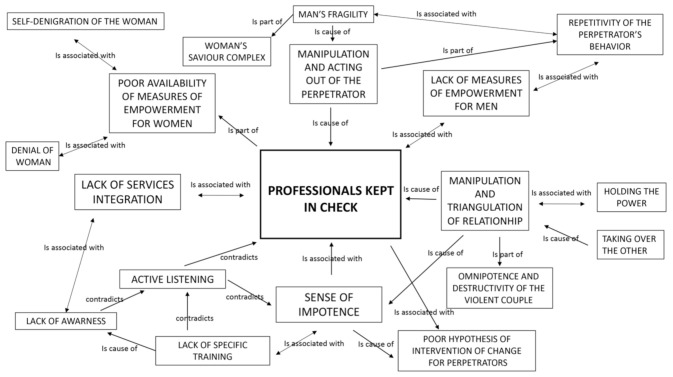
Network and core category “professionals kept in check”.

**Table 1 ijerph-17-07910-t001:** Participants’ data. This table illustrates some information about participants.

Features	Percentages/Frequencies
Gender *n* %	45 F 90
	5 M 10
Professional Role %	62 Psychologists and Psychotherapists 12 Social Workers 12 Lawyers 14 Others
Work Context %	30 Anti-violence center for women 20 OLV (“Oltre La Violenza” project for men) 14 Center for families 36 Others
Years of Service % (range)	12 (1 ≥ 5) 24 (6 ≥ 10) 12 (11 ≥ 15) 52 (>15) Mean 27.5
Years in Dealing with Violence % (range)	32 (1 ≥ 5) 32 (6 ≥ 10) 2 (11 ≥ 15) 34 (>15) Mean 18.31

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
