# Peer review of "“Kept in Check”: Representations and Feelings of Social and Health Professionals Facing Intimate Partner Violence (IPV)"

_ijerph, 2020, doi:10.3390/ijerph17217910_

Round 1

Reviewer 1 Report

The paper deals with a very interesting topic representing a specific facet of the research on IPV. It has a robust sample for a qualitative study.

The introduction is clear. Authors should consider adding some more international references, as they refer mostly to Italian scholars.

A revision from a native English speaker could improve the paper’s clarity. Some sentences (ie. p.2, l. 57-58) are hardly readable and do not make much sense in English (i.e.p.7, l. 309, p. 7,l.  328-329)). The overall paper would benefit form this kind of revision.

The paragraph “1.1. Intimate Partner Violence (IPV): the violent relationship and its consequences” could be better organized, going straightforward to line 73, and clarifying that the first part will discuss the violent relationship and its different interpretations. There is not so much about consequences, differently from what the reader could expect from the paragraph’s title. In any case, this part of the paper should be expanded, as the interpretation of violent relationships is announced as a key theme of the paper. Based on that I would also suggest to describe in more details the studies concerning views of professionals, referring also to a wider literature, not limited to the work done by the authors (see for example the following useful study: Saletti-Cuesta, L., Aizenberg, L., & Ricci-Cabello, I. (2018). Opinions and experiences of primary healthcare providers regarding violence against women: a systematic review of qualitative studies. Journal of family violence, 33(6), 405-420.

The section regarding participants should be reorganized,  clarifying first numbers and professional roles, and then details about demographics.  In order to gain more clarity, I would suggest providing more details about the methodology adopted, and the procedures associated with that method before the description of participants.

The coding procedure is described in abstract and vague terms but does not clarify how authors got to the codes. Moreover given that in the analysis section only some codes are presented it is rather appropriate to explain why they have been selected. It should be a good idea to clarify how did you get to the core category, making the process more explicit, and make stronger connections with previous analysis.

In the conclusions there is a statement about reflexivity, as a key process. However it should be made clear to what extent reflexivity is connected with professionals’ self-awareness about their kept in check position.

I also wonder if there are differences between specific professional profiles regarding the reflexive process and the relevance of the core category. If the authors did not find any specificities it is worth saying it. If there are some, it would be interest to shortly. discuss them 

Author Response

The paper deals with a very interesting topic representing a specific facet of the research on IPV. It has a robust sample for a qualitative study.

1. The introduction is clear. Authors should consider adding some more international references, as they refer mostly to Italian scholars.

THANK YOU, WE HAVE ADDED MORE INTERNATIONAL REFERENCES

2. A revision from a native English speaker could improve the paper’s clarity. Some sentences (ie. p.2, l. 57-58) are hardly readable and do not make much sense in English (i.e.p.7, l. 309, p. 7,l.  328-329)). The overall paper would benefit form this kind of revision.

WE HAVE HAD IT REVISED BY A NATIVE ENGLISH SPEAKER AND SHE HAS CLARIFIED THE ABOVEMENTIONED SENTENCES (e.g. SEE NOW LINES 377-379)

3. The paragraph “1.1. Intimate Partner Violence (IPV): the violent relationship and its consequences” could be better organized, going straightforward to line 73, and clarifying that the first part will discuss the violent relationship and its different interpretations. There is not so much about consequences, differently from what the reader could expect from the paragraph’s title. In any case, this part of the paper should be expanded, as the interpretation of violent relationships is announced as a key theme of the paper. Based on that I would also suggest to describe in more details the studies concerning views of professionals, referring also to a wider literature, not limited to the work done by the authors (see for example the following useful study: Saletti-Cuesta, L., Aizenberg, L., & Ricci-Cabello, I. (2018). Opinions and experiences of primary healthcare providers regarding violence against women: a systematic review of qualitative studies. Journal of family violence, 33(6), 405-420.

THANK YOU. WE HAVE REORGANIZED THE ARTICLE BETTER AND WE HAVE ADDED MORE REFERENCES ON THE CONSEQUENCES OF IPV

4. The section regarding participants should be reorganized, clarifying first numbers and professional roles, and then details about demographics.  In order to gain more clarity, I would suggest providing more details about the methodology adopted, and the procedures associated with that method before the description of participants.

WE HAVE ADDED A TABLE (TABLE 1) DESCRIBING THE PARTICIPANTS’ FEATURES AND WE HAVE INTRODUCED THE METHODOLOGY BEFORE THE “PARTICIPANTS” PARAGRAPH (SEE LINES 174-180 AND TABLE 1 P. 5)

5. The coding procedure is described in abstract and vague terms but does not clarify how authors got to the codes. Moreover, given that in the analysis section only some codes are presented it is rather appropriate to explain why they have been selected. It should be a good idea to clarify how did you get to the core category, making the process more explicit, and make stronger connections with previous analysis.

WE HAVE ADDED MORE DETAILS RELATED TO THE ANALYSIS AND CODING PROCEDURES, MACROCATEGORY AND CORE CATEGORY, AND WE HAVE DESCRIBED THE PROCESS THAT LED TO THE CORE CATEGORY MORE CLEARLY (SEE LINES 223-237)

6. In the conclusions there is a statement about reflexivity, as a key process. However, it should be made clear to what extent reflexivity is connected with professionals’ self-awareness about their kept in check position.

WE HAVE REFORMULATED OUR THOUGHT FROM LINE 511 T0 LINE 513 AND FROM 519TO 524,  CONNECTING THE CONCEPT OF REFLEXIVITY TO PROFESSIONALS’ SELF-AWARENESS ABOUT THEIR KEPT IN CHECK POSITION

7. I also wonder if there are differences between specific professional profiles regarding the reflexive process and the relevance of the core category. If the authors did not find any specificities it is worth saying it. If there are some, it would be interest to shortly. discuss them 

THANK YOU, WE HAVE SPECIFIED THIS POINT FROM LINES 456 TO 461.

Reviewer 2 Report

Overall excellent paper. A lot of good information and this paper can significantly contribute to the field. I have made comments below and have included corresponding numbers to make it easier for the authors to find what section I am referring to.

Feedback for improvement.

48 “full silence of the man”, Not clear what is meant here.

 “Trust” in violence 37 needs to be better defined

This concept needs to be better defined “transversality” 42

49-52 overwhelming women are more subjected to IPV there is a lot of research on that, not just sexual violence.

73-85 argument is problematic. Women stay in relationships for multiple reasons and the way it is written, there is a subtle suggestion of victim blame, as battered women syndrome is also included there.

86-93 is also problematic to me because that suggest violence is beyond control of men. Partner violence is a mostly understood as an act of power and control. I think you have mentioned one side, variations need to be presented.

102-103 is also problematic how women are presented. These are very well research on western countries, such as the US. Your statement suggest victim blaming.

127 spell check-perpetrator.

152 Usually qualitative opened ended interview guides are called semi- structured guides.

341Kept in check- that theme needs to be explained better initially. The concept needs to be first explained in relation to the grounded theory concepts you are establishing.

Finally, I am not sure the diagram you presented is well described. Connected it better to the themes you discussed.

Author Response

Overall excellent paper. A lot of good information and this paper can significantly contribute to the field. I have made comments below and have included corresponding numbers to make it easier for the authors to find what section I am referring to.

Feedback for improvement. 

1. 48 “full silence of the man”, Not clear what is meant here.

WE HAVE SPECIFIED THE MEANING OF THE SENTENCES AT LINES 54 AND 56

2. “Trust” in violence 37 needs to be better defined

IT IS NOT A TRUST IN VIOLENCE, BUT VENTIMIGLIA REFERED TO A TYPE OF VIOLENCE THAT ENTERS INTO  RELATIONSHIPS WHERE FEELINGS LEAD TO RELYING ON THE OTHER WITH TRUST AND SECURITY. (SEE LINES 39-43)

3. This concept needs to be better defined “transversality” 42

WE DEFINED THE TERM “TRASVERSALITY” BETTER, FROM LINE 48 TO LINE 51

4. 49-52 overwhelming women are more subjected to IPV there is a lot of research on that, not just sexual violence.

THANK YOU, WE HAVE ADDED OTHER REFERENCES AND DATA (LINES FROM 59 TO 63)

5. 73-85 argument is problematic. Women stay in relationships for multiple reasons and the way it is written, there is a subtle suggestion of victim blame, as battered women syndrome is also included there.

WE HAVE REFORMULATED THE SENTENCE, AS YOU CAN READ FROM LINE 96 TO LINE 99

6. 86-93 is also problematic to me because that suggest violence is beyond control of men. Partner violence is a mostly understood as an act of power and control. I think you have mentioned one side, variations need to be presented.

 WE HAVE ADDED OTHER ASPECTS, AS YOU SUGGESTED (LINES 109-122)

7. 102-103 is also problematic how women are presented. These are very well research on western countries, such as the US. Your statement suggest victim blaming.

SORRY, WE HAVE REFORMULATED THE CONCEPT. IT WAS NOT OUR INTENTION TO COMMUNICATE A WOMAN'S BLAME, BUT TO SIMPLY UNDERLINE THE STRENGTH AND ROLE OF GENDER STEREOTYPES IN ADHERING TO A CERTAIN TYPE OF RELATIONSHIP (SEE LINES 130-134)

8. 127 spell check-perpetrator.

WE HAVE CORRECTED THIS (SEE NOW LINE 168)

9. 152 Usually qualitative opened ended interview guides are called semi- structured guides.

IN A SEMI-STRUCTURED INTERVIEW QUESTIONS ARE ESTABLISHED BEFORE THE INTERVIEW AND FORMULATED USING THE INTERVIEW GUIDE (MASON 2004, RUBIN & RUBIN 2005, RWJF 2008). QUESTIONS ARE PREDEFINED AND ONLY THE ANSWERS WILL CHANGE ACCORDING TO THE INTERVIEWED VISION. HERE   THE MAIN TOPICS OF THE STUDY ARE COVERED BY THE INTERVIEW GUIDE (TAYLOR 2005). IT SHOULD NOT BE FOLLOWED STRICTLY, BUT REQUIRES  RIGOUR IN THE INTERACTION. IT PROVIDES PARTICIPANTS WITH GUIDANCE ON WHAT TO TALK ABOUT (GILL ET AL. 2008).

OUR INTERVIEW IS DEFINED OPEN AND NOT SEMI-STRUCTURED, BECAUSE IT FOLLOWS DIFFERENT PRINCIPLES, AS DESCRIBED IN THE MENTIONED REFERENCES. THE INTERVIEWER, IN THIS CASE, WITH THE TOPICS OF GENERAL INTEREST IN MIND, FOLLOWS THE FLOW OF THOUGHTS OF THE INTERVIEWED PEOPLE WHO CAN INTRODUCE FURTHER THEMATIC AREAS, NEW MEANINGS ETC. WHICH ARE NOT IGNORED, BUT EXPLORED FURTHER.

10. 341Kept in check- that theme needs to be explained better initially. The concept needs to be first explained in relation to the grounded theory concepts you are establishing.

WE HAVE ADDED SOME LINES TO EXPLAIN THIS POINT BETTER AND INCLUDED THE DECISION THAT LED TO THIS DEFINITION OF THE ISSUE THAT CHARACTERIZED THE RESULTS AND THE FEATURE OF ALL THE ARTICLE (e.g. SEE NOW LINES 412-415)

11. Finally, I am not sure the diagram you presented is well described. Connected it better to the themes you discussed.

THANK YOU, WE HAVE REFORMULATED THE CONCEPTS IN THE DIAGRAM, INDICATING RELATIONSHIP AMONG THEM (SEE FIGURE 1, P. 11).

Round 2

Reviewer 1 Report

 I do not have further specific requests, as the authors have responded to my concerns.